# ESQA: Event Sequences Question Answering

## Abstract

Event sequences (ESs) arise in many practical domains, including finance, retail, social networks, and healthcare. In machine learning, event sequences are a special type of tabular data with annotated timestamps. Despite the importance of ESs modeling and analysis, little effort was made in adapting large language models (LLMs) to the ESs domain. In this paper, we highlight the common difficulties of ESs processing and propose a novel solution capable of solving multiple downstream tasks with little or no finetuning. In particular, we solve the problem of working with long sequences and improve time and numeric feature processing. The resulting method, called ESQA, effectively utilizes the power of LLMs and, according to extensive experiments, achieves state-of-the-art results in the ESs domain.

## 1 Introduction

Temporal data often comes in the form of event sequences, where each event is characterized by the arrival time and additional structured data. This type of data is widely spread in domains like geoscience Bergen et al. (2019), healthcare Esteva et al. (2019), sociology Hossain et al. (2020), industry Choi et al. (2021), e-commerce Ni et al. (2018) and finance Babaev et al. (2022). Event sequences combine properties of time series and tabular data while having major differences. Unlike time series, events can arrive with irregular time steps and have structured annotations, similar to tabular datasets. Unlike tabular data, events have timestamps and associated orders. These differences require special data processing, modeling, and inference approaches.

The new frontier in machine learning, especially deep learning, focuses on adapting large language models (LLMs) to domains beyond language. The reasons behind this adaptation are that LLMs can use additional information not found in domain-specific data, process the textual context of the underlying task, generate answers in a free natural form, argue their decisions, and support dialog with the user. The potential benefits of using LLMs include improved modeling quality and generalization. The latter means that the hybrid model can solve new problems with little or no finetuning, which largely increases the model's applicability and reduces development costs. Successful applications of LLMs were demonstrated in both time series Cai et al. (2023) and tabular datasets Dinh et al. (2022), but no effort was made to adapt LLMs to event sequences: financial transactions, electronic health records, activity on different devices, etc. These data characterize human life and are used to personalize many AI services across different domains.

Event sequence processing with LLM encounters several difficulties. First, structured data must be effectively encoded at the LLM's input. Textual representation considerably increases the sequence length and can't be effectively processed by modern Transformer models due to the quadratic complexity. Second, the desired method must be capable of processing long input sequences, even when the downstream tasks require historical data analysis. The problem is similar to the first but focuses on the model architecture rather than input processing. Finally, time features and the order must be properly provided to the model, as they constitute the essence of event sequences and include important information for solving downstream tasks.

In this paper, we propose a new neural architecture called ESQA that exploits the power of LLMs to model event sequences and solve associated practical tasks. In particular, we, for the first time, develop a question-answering approach with LLM backbone in the event sequences domain. We show the proposed model can solve multiple downstream tasks without finetuning. When finetuned, ESQA outperforms other methods and achieves a new state-of-the-art.

## 2  RELATED WORKS

**Event Sequences.** We assume that events, denoted as $e_i$, are arranged in sequences $S_n = \{e_i\}_{i=1}^{I_n}$ based on their association with a common entity. Here, $I_n$ represents the number of events in the sequence $S_n$. An entity could represent a bank customer or a web user, while the events within the sequence might include actions like a completed transaction or a series of clicks. These events are connected by a temporal order: $t(e_i) < t(e_{i+1})$, where $t(.)$ indicates the time at which the event occurs. Event sequences encompass a diverse range of attributes, with each event, $e_i$, characterized by a set of features $\{c_j\}_{j=1}^{C}$. These features can be depicted as a vector of values with dimension $C$. Additionally, $Y_m$ represents the target variable vector for the problem at hand, which may be based on the value of a sequence feature $c_m$ or external variables, such as a bank client's default status. Event attributes comprise numeric $c_j^{num}$ and categorical features $c_j^{cat}$ of various types. Categorical features define attribute values within a finite set of categories $c_j^{cat} \in |c_j| = \{cat_{j;1}, ..., cat_{j;K_j}\}$, where $K_j$ denotes the number of possible values for the feature $c_j^{cat}$ Lane (2003). Numerical features $c_j^{num} \in \mathbb{R}$ are those represented as numbers, allowing meaningful arithmetic operations to be performed Lane (2003).

Temporal Point Processes (TPPs) and their marked variants (MTPPs) are the simplest forms of event sequences. Previous research was focused on accurate next-event prediction with or without neural networks Liniger (2009); Mei & Eisner (2017); Xue et al. (2023). Another branch of research addressed event streams of the general form Padhi et al. (2021); Babaev et al. (2022); McDermott et al. (2024). To the best of our knowledge, question answering with LLMs was not previously applied to TPPs or event sequence modeling.

**LLMs for Tabular Data.** Large Language Models (LLMs) are a family of neural architectures pretrained on a large corpus of texts. LLMs accept inputs in the form of text and generate textual output. In practice, LLM architecture is composed of three main blocks. The first is an embedding layer that converts input text to a sequence of numeric vectors called embeddings. The second block, the backbone, transforms input embeddings to the output embeddings sequence with possibly different lengths. The final part of the model maps embeddings to the output text.

There are two main approaches for encoding tabular data at the input of LLM. The first one is to provide a description of each table field in the textual form Dinh et al. (2022). This approach suffers from little flexibility and extremely long input sequences. The second approach is to replace the embedding layer, with a newly designed module capable of directly encoding table fields to embeddings with the required number of features. The latter approach is also known as embedding injection and usually achieves better results Koh et al. (2023); Huang et al. (2023).

In our approach, we endeavor to rethink the best practices for making embeddings from categorical and numeric features from tabular neural networks Yin et al. (2020); Iida et al. (2021); Padhi et al. (2021); Hegselmann et al. (2022); Yang et al. (2022); Dinh et al. (2022). At the same time, event sequences require analysis of multiple events simultaneously, while tabular datasets can be processed one row at a time. To this end, ESQA applies the encoding method and adapts Q-Former, which is not seen in tabular neural networks.

**LLMs for time series.** Previous works used LLMs in the context of time series analysis Gruver et al. (2023); Cai et al. (2023); Zhang et al. (2024). ESQA implements a novel context encoding, unlike time series models, and can process complex data structures.

**Question Answering with LLMs.** The popular way to solve problems with LLMs is to design a question such that a valid answer to this question solves the problem Dinh et al. (2022). The question must include the context, i.e. all necessary data for reasoning, and the task definition. This way, LLM input is usually composed of the context, task, and connecting words indicating the boundaries of each part.

## 3  EVENT SEQUENCES QUESTION ANSWERING

The general view of the proposed model, called *Event Sequences Question Answering (ESQA)*, is presented in Figure 1. Below, we will give a detailed description of the model's input and the backbone LLM.

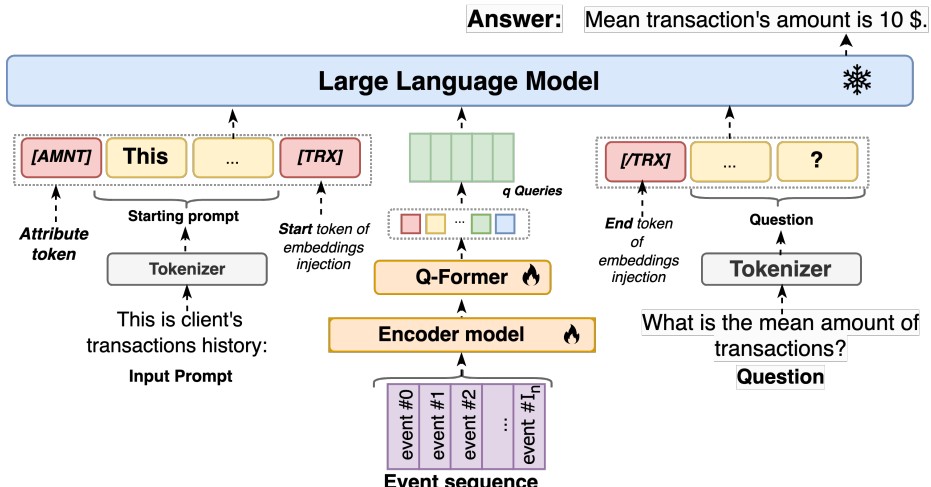

Figure 1: Proposed model architecture. The components of the approach that do not require training are colored in blue. Components whose weights are optimized during training are colored in orange. The trainable embeddings and associated tokens are colored in red.

### 3.1 QUESTIONS AND ANSWERS CONSTRUCTION

The concept behind this method is to frame all tasks involving temporally structured data as natural language questions and answers. Each task from $\{task_m\}_{m=1}^{M}$ takes the form $\{Q_m, X_m, A_m\}$, where $Q_m$ is the question that defines the problem, $X$ represents the input data, and $A_m$ is the answer sought based on the target variable $Y_m$.

A question $Q_m$ consists of two components: the prefix and the question body. The prefix initiates the query token sequence and is placed before embeddings of other modalities. The question body then describes the task in textual form. For example, determining the most frequent value of feature $c_m$ is represented as: *"What is the most frequent value of $c_m$ in the entire dataset?"*.

Given the nature of time-structured data, we classify questions into extractive and predictive. Extractive questions focus on tasks involving existing event sequences, such as computing statistics or identifying trends and characteristics. On the other hand, predictive questions pertain to tasks concerning predicting future events or attributes based on available data.

Tasks and their corresponding questions can also be categorized based on the type of response sought: binary, multiple-choice, or open-ended. Binary questions seek a straightforward answer, either as $A_m \in \{0, 1\}$ or in the form of *"Yes"* or *"No"*. For instance, a question like *"Is drinking water the most frequently purchased product?"* can be answered with a simple *"Yes"* or *"No"*.

In contrast to binary questions, multiple-choice and open-ended questions assume a specific answer corresponding to the required feature, whether numerical $A_m \in \mathbb{R}$ or categorical $A_m \in |c_j| = \{cat_1, \ldots, cat_K\}$. Multiple-choice questions provide a list of possible answer choices. For example, one might ask *"What is the most frequently purchased product? Options: black tea; bread; drinking water; grapes."*. On the other hand, open-ended questions prompt a direct response, such as *"What is the name of the most frequently purchased product? Please provide the name in your response."*.

### 3.2 EVENTS EMBEDDINGS

To address the integration of event sequences into a language model, we propose adapting the method outlined in previous works Koh et al. (2023); Huang et al. (2023). This involves embedding multi-modal information into an LLM, parameterized by $\theta$, directly mapping it into the intrinsic embedding space $E^\theta$, bypassing the discrete text token layer. To achieve this, we introduce a trainable mapping $\phi : Z \to E^\theta$, where $Z$ represents the observation space of temporally structured data. This mapping converts the data into a sequence of $f$-dimensional vectors in $E^\theta$, which are then integrated

into a sequence of text embeddings. This interleaving of modalities creates a multi-modal input for the LLM.

ESQA represents all event features as trainable embeddings. It is achieved by encoding each value $x_{ij}$ of a categorical or integer numeric feature $c_j$ with a sequential index $k_{x_{ij}}$ based on the total number of unique values for that feature $k = [0, \ldots, K_j]$. This index uniquely identifies the embedding $emb_k$ of a feature value in the embedding matrix $W_e$. The embedding dimension is selected based on the formula: $dim(e_k) = \lceil \lambda \times K_j^\mu \rceil$. The coefficients $\lambda = 1.6$ and $\mu = 0.56$ have been chosen empirically.

Numerical features in the form of real numbers are discretized into non-overlapping intervals: $B_j^1, \ldots, B_j^n$, $B_j^i = [b_j^{i-1}, b_j^i)$, where $j$ - is an index of numeric features. The distribution of the feature $c_j$ in the training sample is used to determine these intervals. The number of intervals is chosen based on the approach in Doane (1976), using the formula $n = 1 + \log_2(n) + \log_2(1 + \frac{|g_1|}{\sigma_{g_1}})$, where $g_1$ is the estimated third-moment skewness of the distribution and $\sigma_{g_1} = \sqrt{\frac{6(n-2)}{(n+1)(n+3)}}$. This method is particularly suited for distributions of features that deviate significantly from the normal distribution. Once the intervals have been defined, the value $x_{disc}^{num}$ of $j$'th numerical feature is defined as follows:

$$x_{disc}^{num} = \begin{cases} b_j^0, & x_{ij} < b_j^0, \\ b_j^n, & x_{ij} \geq b_j^n, \\ b_j^i & b_j^{i-1} \leq x_{ij} < b_j^i. \end{cases} \tag{1}$$

In addition to this approach to the vectorization of numerical and temporal features, a comparative analysis of existing encoding approaches has been conducted. The results of this analysis are presented in Section A.2.1.

The resulting feature embeddings are concatenated into a tensor $e_i^{emb}$ of dimension $dim(e_i^{emb}) = \sum_{j=1}^{C} |c_j|$, which describes a single event $e_i$ from the sequence. A vector representation of sequence $S_n$ is formed by combining vector representations of individual events into a joint tensor $S_n^{emb}$ shown in Fig.2a.

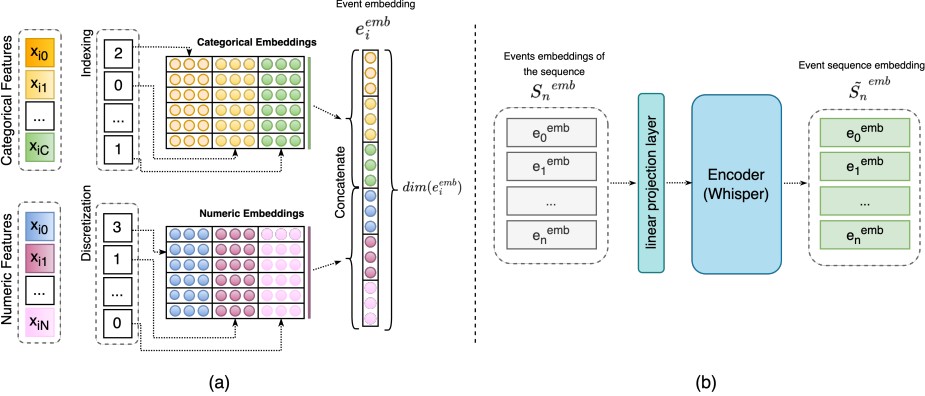

Figure 2: a) Event sequences feature encoding; in the example, there are $N$ numerical and $C$ categorical features, which are concatenated into a tensor $e_i^{emb}$ of dimension $dim(e_i^{emb})$. b) The event sequence encoder model processes the concatenated feature embedding vectors $S_n^{emb}$ for all events within a sequence, ultimately producing a comprehensive embedding $\tilde{S}_n^{emb}$ for the entire event sequence.

### 3.3 ENCODER

After the initial layer of input data embeddings, vectorized event sequences are fed into a specialized encoder model Fig. 2b. This module, based on the architecture of the Transformer decoder, processes sequences of events in an autoregressive manner by predicting each subsequent event. For our implementation, we used both Whisper-tiny and Whisper-small models Radford et al. (2022), initialized with weights pre-trained on audio data. The input tensor for the encoder comprises concatenated feature embedding vectors for all events $S_n^{emb}$ (Section 3.3.1) and has a size of $dim(S_n^{emb}) = (I_n, dim(e_i^{emb}))$. The encoder processes this tensor autoregressively, similar to the sequence of text token embeddings, resulting in a sequence of vectors $\tilde{S}_n^{emb}$ with a size $dim(\tilde{S}_n^{emb}) = (I_n, d_{enc})$. Here, $d_{enc}$ represents the output layer dimensionality of the encoder model. To ensure compatibility between the dimensions of the input embeddings of the event sequences $dim(S_n^{emb})$ and the embedding layer of the encoder model $d_{enc}$, we used a linear projection layer.

This choice of encoder architecture is motivated both by the temporal nature of the event sequences, which aligns with autoregressive modeling and by the results of a series of experiments. Appendix A.1 provides a detailed description of the experiments and their results.

### 3.4 CONNECTOR

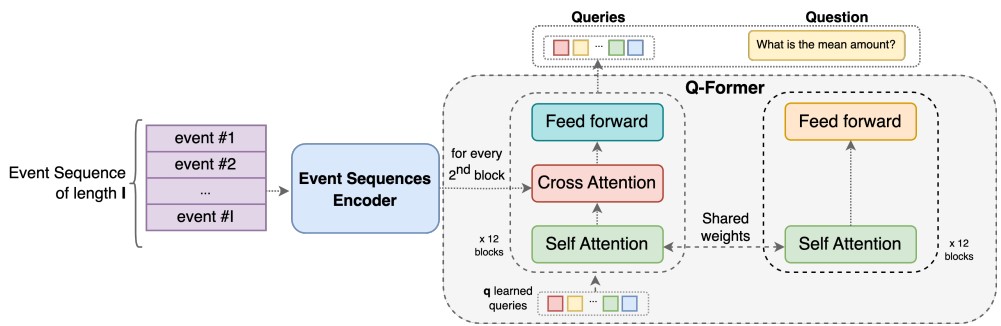

Figure 3: The Q-Former model's architecture is designed to extract the most relevant event sequence representations. It produces $q$ query embeddings for each event sequence, which are then linearly projected to the size of the language model embedding and appended to the embedded question tokens. Subsequently, the joint sequence is transmitted to the LLM.

The output representation of the event sequence encoder grows in dimensionality as the number of events in each sequence increases. This size is crucial, as it must fit within a common multimodal embedding sequence, impacting the language model's context length extension. Our goals are to shorten the event sequence length without significant information loss and to adapt each event's vectorized representation to match the language model's embedding dimension. We propose an intermediate connection layer between the event sequence encoder and the LLM to achieve this. We suggest using the Query Transformer model, or Q-Former Li et al. (2023), to extract features from the encoder output efficiently.

The Q-Former architecture (Fig. 3) includes two transformer submodules: a novel modality transformer (originally an image transformer) that works with a fixed image encoder for feature extraction, and a text transformer that functions as both an encoder and a decoder. A set of trainable query embeddings $q$ is the input for the novel modality transformer. These queries engage in self-attention, interacting with each other and with the fixed modality features through cross-attention layers in every other transformer block.

In our approach, Q-Former produces $q$ query vectors for each event sequence, which are then passed to the LLM. We use a single fully connected layer to project the output query vectors into the language model's text embedding dimension. In this study, we initialize Q-Former with the weights from the BLIP-2 approach, derived from BLIP-2 with the FLAN-T5-xl model Li et al. (2023). The

architecture and initialization of the connection layer were chosen based on a series of experiments detailed in Appendix A.2.

## 3.5 TRAINING PROTOCOL

As the backbone for the pre-trained LLM, our approach utilizes the FLAN-T5 family of encoder-decoder models Wei et al. (2021). Any process of fine-tuning model parameters influences the model's proficiency in a specific domain and causes it to "forget" essential general and linguistic knowledge. We have frozen most of the LLM parameters to preserve this knowledge and save computational resources. Studies Lu et al. (2021); Zhou et al. (2023) indicate that freezing most of the model's weights often yields better results than fully fine-tuning a pre-trained LLM.

To efficiently select a limited set of trainable parameters, we propose using Parameter-Efficient Fine-Tuning (PEFT) methods. Specifically, we employed the Low-Rank Adaptation (LoRA) approach Hu et al. (2021), which keeps most of the model weights frozen while adding trainable rank decomposition matrices to a subset of the parameters.

Similar to the conventional language model training process, the ESQA approach was trained to minimise the cross-entropy loss function between the predicted probability distribution of text tokens and the predefined tokens of ground-truth answers.

The training of the ESQA approach to solve multiple problems is conducted in a single stage, during which the LLM weights are maintained frozen and the following components are optimised:

- categorical, numeric and temporal features embeddings;
- an event sequences encoder;
- a connector as a Q-Former model;
- a linear layer, that is responsible for mapping queries vectors from the connector to the input embedding space of LLM;
- decomposition matrices $W^A \in \mathbb{R}^{r \times hidden\ size}$ and $W^B \in \mathbb{R}^{hidden\ size \times r}$ for $W_q$ and $W_v$ matrices of the self-attention and encoder-decoder attention layers.

## 4 EXPERIMENTS

In this section, we begin by presenting the evaluation details, which include the comparison methods and the datasets used for evaluation. Following this, we conduct a series of systematic experiments to showcase the capability of the developed ESQA approach in addressing a diverse range of problems based on event sequences.

### 4.1 EXPERIMENTAL SETUP

**Datasets.** Sequences of events are prevalent across various domains and tasks, with a particularly high demand for data analysis in the fintech sector. In this field, the transactional activity of individuals serves as the primary source of information. Consequently, we have chosen to utilize a collection of datasets containing customer transactions from banks and marketplaces as examples of event sequences. Given the limited availability of public datasets in this area, the sensitivity of the information in these datasets has significantly influenced our choice and the number of datasets used.

We selected five publicly available datasets featuring sequences of bank transactions: AlfaBattle 2.0, the Age Group Prediction Competition, the X5 Retail Hero dataset for Uplift Modeling in Promotional Campaigns, the Gender Prediction Competition using the evaluation benchmark described in Babaev et al. (2022). In addition, we used the Taiwan Default of Credit Card Customers dataset from Yeh & hui Lien (2009) with short sequences of events. A comprehensive description of the datasets can be found in Appendix B.2.

**Training and hyperparameters.** All experiments for ESQA described below utilized consistent hyperparameters and approach components unless otherwise specified. We employed the AdamW optimizer Loshchilov & Hutter (2017) with parameters $\beta_1 = 0.9$, $\beta_2 = 0.98$, and a weight decay

Table 1: Hyperparameters used for ESQA training. In all experiments, the Whisper-small model architecture was used as the encoder.

| Dataset | AlfaBattle2 | Age | Gender | X5 | Taiwan |
|---|---|---|---|---|---|
| LLM | flan-T5-xl | FLAN-T5-xl | fla-T5-large | FLAN-T5-xl | FLAN-T5-xl |
| Emb. size | 201 | 110 | 74 | 163 | 100 |
| Learn. rate | 3e-4 | 3e-4 | 1e-4 | 1e-4 | 1e-4 |
| warmup steps | 4 k. | 1 k. | 1 k. | 4 k. | 1 k. |
| Max. epochs | 40 | 10 | 10 | 20 | 30 |
| Batch size | 300 | 250 | 50 | 250 | 50 |
| Min seq. len. | 50 | 0 | 0 | 0 | 6 |
| Max seq. len. | 750 | 1500 | 1500 | 750 | 6 |

Table 2: Trainable parameters of LLM with LoRA.

| Model | % trainable params. | # trainable params. |
|---|---|---|
| FLAN-T5-small | 0.8862 | 0.688 M. |
| FLAN-T5-base | 0.7096 | 1.77 M. |
| FLAN-T5-large | 0.5989 | 4.72 M. |
| FLAN-T5-xl | 0.3301 | 9.44 M. |
| FLAN-T5-xxl | 0.1692 | 18.87 M. |

of 0.01. Cosine learning rate decay with restarts was applied, featuring different peak learning rates for each dataset and varying numbers of warm-up steps. In our experiments, LoRA Hu et al. (2021) with a rank of $r = 16$ was applied only to the matrices $W_q$ and $W_v$ of the self-attention and encoder-decoder attention layers. The LoRA scaling factor was set to 32, and the dropout rate to 0.05. The number of trainable parameters in the language model was calculated as $\theta^{train} = 2 \times L \times d_{model} \times r$, where $L$ is the number of layers and $d_{model}$ is the internal dimensionality of the language model. The rank of trainable decomposition matrices is denoted by $r$. Therefore, the number of trainable parameters in each FLAN-T5 model did not exceed 0.9% of the total parameters (Table 2). All models were trained using 6 Nvidia A100 (80G) GPUs. The training hyperparameters are summarised in Table 1.

**Baselines.** We selected representative baseline approaches for analyzing selected transactional datasets, which have proven effective across various benchmarks. Most baseline methods are detailed in Babaev et al. (2022). Additionally, Padhi et al. (2021) presents an implementation of the state-of-the-art transformer architecture for multivariate time series, which closely resembles transactional data. Recent work by Skalski et al. (2023) proposed a new method for transactional data that achieved outstanding results on the AlfaBattle and Age datasets. To ensure a comprehensive evaluation, we adopted a more thorough evaluation protocol from the repository of the paper Babaev et al. (2022), reapplying it to models and datasets not covered in the CoLES repository. Further details on the selection of baselines and their relevance to our study can be found in Appendix B.3.

For the next event prediction task, we also provide calculated statistical baselines for numerical and categorical target features to compare the quality of prediction tasks in zero-shot setups. These include, for example, predicting the mean or median value for numerical target variables and the most frequent value for categorical attributes.

A complete list and detailed description of baselines are provided in Appendix B.3.

## 4.2 EXPERIMENTAL RESULTS

### 4.2.1 MAIN RESULTS

Each dataset used for model evaluation corresponds to a specific downstream task. For instance, the AlfaBattle dataset predicts a bank customer's loan default, while the Age dataset is employed to predict the age group. It is important to highlight that the AlfaBattle dataset is highly imbalanced, with the positive class constituting less than 3%, while the Gender dataset has a slight over-representation

Table 3: A comparison of ESQA on the downstream tasks of the five event sequence datasets described in Section 4.1 with the baseline approaches presented in Section 4.2.1. The best results are highlighted in **bold** and the second best results are underlined.

| Dataset | AlfaBattle | Age | Gender | X5 | Taiwan |
| --- | --- | --- | --- | --- | --- |
| Metric | AUCROC | Accuracy | AUCROC | Accuracy | AUCROC |
| Handcrafted feat. | 0.7792 | 0.629 | 0.877 | 0.547 | 0.784 |
| Randomly init. RNN | 0.6456 | 0.375 | 0.593 | 0.368 | 0.722 |
| CPC | 0.7919 | 0.602 | 0.851 | 0.525 | 0.732 |
| Barlow Twins | 0.7878 | 0.634 | 0.865 | 0.521 | 0.611 |
| CoLES | 0.7921 | 0.640 | **0.881** | 0.539 | 0.716 |
| NSP | 0.7655 | 0.621 | 0.852 | 0.425 | 0.670 |
| RTD | 0.7910 | 0.631 | 0.855 | 0.520 | 0.675 |
| SOP | 0.7238 | 0.512 | 0.785 | 0.428 | 0.781 |
| TabFormer | 0.7862 | 0.580 | 0.828 | 0.393 | 0.679 |
| GPT | 0.7737 | 0.574 | 0.785 | 0.511 | 0.732 |
| NPPR | **0.7980** | 0.642 | - | - | - |
| **ESQA (ours)** | 0.7568 | **0.699** | 0.850 | **0.598** | **0.793** |

of the positive class. Therefore, we used the ROC-AUC metric for problems with binary target variables and class imbalance. For multiclass classification with balanced classes, we employed Accuracy. A more detailed explanation of the metric calculation methodology and the assessment of response quality for ESQA is provided in Appendix B.1.

The results of the experiments on the downstream tasks for datasets described in Section 4.1 are summarized in Table 3. The results indicate that the ESQA approach matches both self-supervised contrastive and supervised methods in quality. Notably, on the Age and X5 datasets, ESQA surpasses the baseline scores. Although specific comparative results for other models are unavailable for the Taiwan dataset, ESQA's impressive performance underscores its effectiveness. These outcomes highlight ESQA's superior capability in handling multi-class classification tasks with balanced classes.

However, the results for the client default problem on the AlfaBattle dataset and the Gender dataset are less clear-cut. The CoLES contrastive approach achieves the highest quality for these problems. While ESQA slightly lags behind CoLES, it still shows competitive performance, closely following models like Barlow Twins and RTD and outperforming the SOP approach. It is important to note that both datasets exhibit class imbalance, which is especially pronounced in the AlfaBattle case.

This leads us to conclude that the ESQA approach performs classification tasks as well as, or better than, the selected baseline methods. However, it is significantly affected by the imbalance of the target variable. This limitation can be attributed to the nature of LLMs, originally designed to extract common patterns from text data to model complex language structures.

### 4.2.2 PREDICTIVE TASKS

Most tasks involving event sequences require answering predictive questions about event features. To address such challenges, we propose utilizing the ESQA approach in a multi-task setting, enabling simultaneous predictions of all features of the next event in the sequence. Experimental results for predictive questions against baselines are detailed in Table 4.

On categorical feature prediction tasks, such as MCC code attribute prediction, ESQA achieves the highest performance with an Accuracy/F1 scores, outperforming all other models, with the closest being CPC. This indicates that ESQA is particularly effective in handling categorical prediction tasks within the context of transaction history.

While the Text LLM achieves the lowest MAE/MSE in predicting the numerical amount attribute, ESQA still performs competitively. Although ESQA is not the top performer here, it maintains reasonable accuracy, demonstrating its versatility across different prediction tasks.

Table 4: Table comparing ESQA with the baseline approaches presented in Section 4.2.1 for predicting attributes of the next transaction on the AlfaBattle dataset. The best results are highlighted in **bold** and the second best results are underlined.

| Attribute | MCC code | Amount | Hour diff |
|---|---|---|---|
| Metric | Acc./F1 | MAE/MSE | MAE/MSE |
| CoLES | 0.440 / 0.351 | 0.197 / 0.082 | 36.05 / 1586.52 |
| CPC | 0.475 / 0.411 | 0.196 / 0.074 | 34.89 / 1508.71 |
| RNN with CoLES | 0.469 /0.411 | 0.184 /0.077 | 32.25 / 1573.02 |
| CatBoost | 0.440 /0.367 | 0.190 /0.090 | 34.40 / 1613.41 |
| GPT with descr. | 0.462 /0.423 | 0.179 / 0.083 | 32.63 / 1726.42 |
| Text LLM | 0.382 / 0.381 | **0.103 / 0.0176** | 116.38 / 62161 |
| ESQA (ours) | **0.546 / 0.546** | 0.191 / 0.1021 | **18.313 / 1033.87** |

Table 5: Table comparing the generalisation abilities of the ESQA approach with the statistical baseline approaches presented in Section 4.2.1, and a text-based approach. The ESQA approach trained on predictive tasks in a multitask setting is referred to as 'ESQA m/t'. While ESQA trained on contextual tasks and adapting to new tasks is referred to as 'ESQA z/s'. The best results are highlighted in **bold** and the second best results are underlined.

| Attribute | Stat. baseline | Text-only | ESQA m/t | ESQA z/s |
|---|---|---|---|---|
| MCC code, acc. | 0.388 | 0.382 | **0.546** | 0.381 |
| MCC category, acc. | 0.437 | 0.402 | **0.588** | 0.435 |
| Amount, MAE/MSE | 0.241 | **0.103/0.018** | 0.191/0.102 | 0.389/0.228 |
| City, acc. | 0.704 | 0.691 | **0.731** | 0.343 |
| Country, acc. | 0.970 | 0.970 | **0.972** | 0.971 |
| Currency, acc. | 0.987 | 0.986 | 0.987 | **0.988** |
| Op. type gr., acc. | 0.766 | 0.733 | **0.840** | 0.781 |
| Op. type, acc. | 0.499 | 0.393 | **0.633** | 0.543 |
| Op. kind, acc. | 0.548 | 0.494 | **0.693** | 0.598 |
| Days before, MAE/MSE | 140.5 / 23823.3 | 10.5 / 657.2 | **6.3/195.9** | 11.394 / 666.2 |
| Hour diff, MAE/MSE | 36.33 | 116.4/62161 | **18.3/1033.9** | 48.85/3980 |

For the temporal Hour diff attribute, ESQA significantly outperforms all other models. The next best model, RNN CoLES, has a much higher MAE/MSE, highlighting ESQA's superior capability in handling temporal prediction tasks effectively.

### 4.2.3 GENERALIZATION ABILITIES

LLMs possess an extraordinary capacity to generalize to novel, previously unseen tasks. Our method maintains the integrity of the language model's weights, thereby preserving its inherent capabilities. Moreover, by training adaptors within the attention layers, we expand the domain of zero-shot tasks from exclusively text-based tasks to those based on event sequences. Following comprehensive pre-training on contextual tasks, we evaluated the ESQA approach's adaptability to new predictive tasks. The model was trained in a multi-task setting on all event features of the AlfaBattle dataset and was subsequently tested in a zero-shot setting across various predictive tasks within the same dataset. Table 5 compares our experimental results against statistical baselines, a text baseline, and an ESQA model specifically trained on those predictive tasks.

For the MCC code and MCC category attributes, ESQA multi-task outperforms all baselines, indicating its strength in handling categorical predictions. However, in the zero-shot setting, ESQA's performance is comparable to the statistical baseline, suggesting room for improvement in scenarios without task-specific training. In predicting the Amount attribute, the text-only approach achieves the best MAE/MSE, while ESQA multi-task shows competitive performance, demonstrating its robustness in handling regression tasks despite not being the top performer. However, the regression

problem on real numbers with many decimals is still a challenging task for zero-shot ESQA, which performed poorly. For temporal predictions like Days before and Hour diff, ESQA multi-task significantly outperforms other approaches, showcasing its superior capability in modeling temporal patterns. Overall, ESQA zero-shot performance, while not leading, still provides valuable insights into ESQA's versatility and potential for improvement in less customized settings.

## 5    CONCLUSION

In this paper, we introduced Event Sequences Question Answering (ESQA), a novel approach for modelling event sequences with LLMs. Our empirical results demonstrate that our approach performs robustly across various datasets. For several downstream problems, ESQA performs at least as well as specialized baselines (Table 3), and for the task of predicting the attributes of the next event, it significantly surpasses baseline methods (Table 4). Furthermore, we have shown that ESQA can handle multiple tasks simultaneously without any special fine-tuning (Table 5), highlighting its remarkable ability to adapt swiftly to new tasks without the need for complex and time-consuming training. These findings position ESQA as an exceptionally promising approach leveraging the strong generalisation capabilities of LLM backbones for the field of event sequences. The source code will be made publicly available at: https://anonymous.4open.science/r/ESQA-AD2A.

**Limitations.** Our research has certain limitations. In processing numerical features, ESQA employs value discretization, which introduces an inherent discretization error. This error is significantly influenced by the number of discretization buckets and the ranges of the actual feature values. To mitigate this error, we conducted several additional experiments to refine the pre-processing method. Furthermore, handling time features in event sequences requires special attention. We are actively exploring ways to enhance temporal feature processing within ESQA. In future work, we will focus on implementing these improvements and addressing the challenge of dealing with unbalanced classes.

**Ethical Statement.** The purpose of this paper is to advance the field of event sequence analysis by leveraging the extensive capabilities of Large Language Models (LLMs). We propose that integrating classical approaches to event sequence analysis with LLMs represents a promising research direction. Such integration has the potential to significantly enhance both practical decision-making and analytical intelligence in the domain.

While our primary focus is on contributing to the academic landscape, our research also has potential societal implications, particularly in decision-making processes and user analytics across various error-sensitive industries. From an ethical standpoint, the responsible and transparent application of LLMs requires a thorough understanding of their strengths and limitations, acknowledging the possibility of errors without intent to cause harm. At this stage of the research, we do not foresee immediate social risks that warrant special attention. However, we emphasize the need to continually assess ethical considerations as this interdisciplinary field evolves and its impact on society becomes clearer.

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

## A  ARCHITECTURE COMPONENTS SELECTION

### A.1  EVENT SEQUENCE ENCODER ARCHITECTURE SELECTION

Training a language model to understand a different modality is not a novel challenge and has been addressed for various data types. Therefore, in our experiments on the event sequence encoder architecture, we built upon advancements from other modalities. We focused on established models for three highly developed modalities: text, images, and audio. For text architectures, we examined several models including encoder-only models like BERT (base, large), encoder-decoder models like T5 (small, base, large), and decoder-only models like GPT (base, medium, large). For image architectures, we used ViT (base, large). For audio models, we considered various versions of Whisper (tiny, small, medium), utilizing only the decoder part of the Whisper architecture.

The models were compared based on their ability to predict the default of a bank client in the AlfaBattle 2.0 dataset, a binary problem where the task is to determine if a bank client will repay a loan based on their transaction history over two years. We used AUC as the metric for comparison. All models were trained from scratch.

We maintained a consistent training scheme across all experiments, employing Adam as the optimizer with a learning rate of 1e-4. A linear warm-up of the learning rate was applied for the first

epoch, followed by a linear decay to zero after 10 epochs. To ensure compatibility between the dimensions of transaction embeddings and the dimensions of pretrained model embeddings, we used a linear layer for text and audio models. Since ViT models cannot process sequences, we addressed this issue by applying a single layer of cross-attention to a fixed number of learnable latent tokens.

As shown in Table 6, decoder-only models outperformed both encoder-only and encoder-decoder models in event sequence encoding in almost all setups. Specifically, experiments with text models demonstrated that the decoder-only GPT2 model outperformed the encoder-decoder T5 model, and the BERT model training did not converge. Similarly, audio architectures, which are primarily decoder-based, also showed superior performance. In response to the concerns about the performance of larger models within the same family, as observed in Table 6, our analysis suggests that the enlargement of the encoder size contributes to overfitting. This overfitting is the primary reason for the degradation in performance outcomes.

Table 6: Table comparing different architectures for predicting default of the client on the AlfaBattle dataset. The best results are highlighted in **bold** and the second best results are underlined.

| Architecture | Type | Number of parameters | AUC |
|---|---|---|---|
| GPT2 Base | Decoder | 124M | 0.7869 |
| GPT2 Medium | Decoder | 355M | 0.7833 |
| GPT2 Large | Decoder | 774M | 0.7747 |
| Whisper-tiny | Decoder | 29M | 0.7892 |
| Whisper-small | Decoder | 153M | **0.7894** |
| Whisper-medium | Decoder | 456M | 0.7715 |
| T5 Small | Encoder-Decoder | 60M | 0.7721 |
| T5 Base | Encoder-Decoder | 223M | 0.7756 |
| T5 Large | Encoder-Decoder | 770M | Diverged |
| BERT Base | Encoder | 110M | Diverged |
| BERT Large | Encoder | 335M | Diverged |
| ViT Base | Encoder | 85M | 0.7822 |
| ViT Large | Encoder | 302M | 0.7639 |

After determining the type of architecture (i.e., the decoder), we conducted further experiments to identify the specific type and size of the decoder architecture. We compared Whisper-tiny, Whisper-small, and GPT2-base, as they produced the best results. Additionally, we evaluated various types and sizes of recurrent architectures: GRU-1, GRU-6, GRU-12, LSTM-1, and LSTM-4, where the number indicates the number of layers used in each model. The embedding size for all recurrent models was set to 1024.

Table 7: Table comparing different **decoder** architectures presented in Section A.1 for predicting default and attributes of the next transaction on the AlfaBattle dataset. The best results are highlighted in **bold** and the second best results are underlined.

| Architecture | # params. | Amount | MCC Category | 24-hour acc | Default |
|---|---|---|---|---|---|
| Metric | | MSE | Accuracy | Accuracy | AUC |
| Whisper-tiny | 29 M. | 0.0660 | 0.4861 | Diverged | 0.7892 |
| Whisper-small | 153 M. | **0.0656** | **0.4896** | 0.645 | **0.7894** |
| GPT-2-base | 100 M. | 0.0657 | 0.4888 | Diverged | 0.7869 |
| GRU-1 | 0.3 M. | 0.0668 | 0.4817 | 0.418 | 0.7854 |
| GRU-big | 16 M. | 0.0670 | 0.4805 | Diverged | 0.7578 |
| GRU-large | 35 M. | 0.0662 | 0.4815 | Diverged | 0.7732 |
| LSTM-1 | 0.4 M. | 0.0669 | 0.4830 | 0.634 | 0.7710 |
| LSTM-4 | 2 M. | 0.0664 | 0.4858 | **0.655** | 0.7664 |

Table 7 indicates that transformer architectures outperformed recurrent models. Scaling up recurrent models did not significantly enhance their quality and sometimes even degraded their performance. Given the similar results among transformer architectures, we selected Whisper-small as the optimal model for all ESQA experiments.

Table 8: Table comparing different **connector** architectures for better modalities alignment. Q-Former architecture based connectors with initialisations from BLIP-2 (Li et al., 2023) pretrained weights are labelled 'w. init.', without initialisation are indicated by 'w/o. init.'. The best results are highlighted in **bold** and the second best results are underlined.

| Architecture Metric | # params. | MCC code Accuracy | MCC category Accuracy | Amount MSE |
|---|---|---|---|---|
| Linear | 197 k. | 0.501 | 0.574 | 0.0174 |
| 2 x Linear | 920 k. | 0.523 | 0.561 | 0.0196 |
| RNN (LSTM) | 3.94 M. | 0.509 | 0.558 | 0.0220 |
| Transformer | 1.1 M. | 0.478 | 0.529 | 0.1361 |
| 2 x Transformer | 2.09 M. | 0.519 | 0.555 | 0.0168 |
| Q-Former-small | 14.7 M. | 0.519 | **0.579** | **0.0162** |
| Q-Former-base (w/o init.) | 96 M. | 0.526 | 0.570 | 0.0189 |
| Q-Former-base (w. init.) | 96 M. | **0.527** | 0.569 | 0.0177 |

## A.2 CONNECTOR ARCHITECTURE SELECTION

Integrating multiple modalities within a single approach centered around an LLM requires mapping new modalities into a textual model. Employing a separate encoder for each modality simplifies the task to finding an efficient architecture for mapping each modality's vector space to the LLM embedding text space. When analyzing event sequences, processing extended data sequences presents challenges due to increased context length, which leads to higher computational complexity. In some instances, the sequence length may surpass the maximum context length of the language model.

To address these challenges, we conducted experiments to determine the optimal architecture for the connection layer between the event sequence encoder and the LLM. We evaluated several potential implementations: a single linear layer, a transformer layer, and two model sizes of the Q-Former architecture. Additionally, we investigated the impact of initialization on problem-solving quality and training speed by initializing the Q-Former with weights from the pre-trained visual-text model BLIP-2, based on FLAN-T5. In all experiments, we tackled three tasks in a multi-task mode using the AlfaBattle dataset. The components used in all experiments included Whisper-tiny as the transaction encoder and FLAN-T5-small as the language model. Performance was measured at 20 epochs, with fixed batch size, learning rate, and optimization parameters. We used multi-class accuracy for classification tasks and MSE for numerical response prediction tasks as target metrics.

The results revealed that simply increasing the number of trainable parameters does not necessarily enhance task solution quality. A linear layer with a small number of parameters performed worse than Q-Former-small, which also trained much faster. However, adding more simple identical blocks within a single connector, such as $'2xLinear'$, did not significantly improve performance. On the other hand, more complex blocks, such as $'2xTransformer'$, showed substantial quality gains. Increasing the model size to Q-Former-base yielded mixed results: while MCC code prediction quality improved by 2%, the metrics for MCC category prediction and numerical attribute Amount declined.

Additional initialization with weights from visual-text pre-training marginally improved the MCC code prediction task but slightly degraded the metrics for the other two tasks. The overall impact of initialization was minimal, indicating few common patterns between extracting salient information from images and deriving dependencies from event sequences. This discrepancy is expected due to the lack of temporal dependence within a single image, in contrast to the strong temporal dependence between events in a sequence.

Therefore, we selected the Q-Former-base model without initialization, anticipating an increase in the number of tasks our approach can handle simultaneously. This model offers a sufficient margin for increasing the complexity of future experiments.

Table 9: The results of comparing different encoding schemes for numerical and temporal features. The estimated metrics are MAE/MSE for both tasks. The best results are highlighted in **bold**.

| Numeric embeddings type | Input features type | Amount | Hour diff |
|---|---|---|---|
| Linear | Raw features | **0.0822 / 0.0135** | 25.8 / 1886.3 |
| Linear | Discretized of features | 0.0822 / 0.0148 | **24.6 / 1775.7** |
| Piecewise | Raw features | 0.099 / 0.0169 | 45.2 / 3640.6 |
| Periodic | Raw features | 0.1011 / 0.0178 | 45.1 / 3638.8 |
| Periodic | Discretized features | 0.0903 / 0.0147 | 30.8 / 2339.0 |

### A.2.1 ENCODING OF NUMERIC AND TEMPORAL FEATURES

In our experiments, we considered different schemes for encoding numerical and temporal features to represent them more efficiently. In particular, we explored three types of techniques suitable for representing real-valued features.

The first is the conventional linear representation of real numbers, implemented by a trainable linear layer. In this case, we also considered two variants of feature input: 1) feeding the original, unaltered feature values, and 2) discretization into a predefined number of intervals, as a technique well established in machine learning Dougherty et al. (1995).

The second approach is piecewise linear encoding, which produces alternative initial representations for the original scalar values and is based on feature binning, a long-established preprocessing technique Gorishniy et al. (2022). Piecewise linear encoding (PLE) relies on a numerical encoding scheme given in Eq. 2, where $B_j^1, \ldots, B_j^n$, $B_j^i = [b_j^{i-1}, b_j^i)$ - non-overlapping discretization intervals, n - number of intervals for $j$'th numerical feature $x^{num}$.

$$PLE(x^{num}) = [e_1, ..., e_n] \in \mathbb{R}^n$$

$$e_i = \begin{cases} 0 & \text{if } x^{num} \leq b_j^{i-1} \text{ and } i > 1 \\ 1 & \text{if } x^{num} \geq b_j^i \text{ and } i < n \\ \frac{x^{num} - b_j^{i-1}}{b_j^i - b_j^{i-1}}, & \text{otherwise} \end{cases} \quad (2)$$

The third approach relies on periodic activation functions Gorishniy et al. (2022) to encode both numerical and temporal features as shown in Eq. 3, where $c_i$ are trainable parameters.

$$Periodic(x^{num}) = concat[sin(v), cos(v)], \quad v = [2\pi c_1 x^{num}, ..., 2\pi c_k x^{num}] \quad (3)$$

The experimental results of the above mentioned techniques are summarised in Table 9. In order to better demonstrate the encoding methods for both numerical and temporal attributes, which are represented as numerical values, we chose for comparison two tasks from the AlfaBattle 2.0 dataset for predicting the real-valued feature of the next transaction Amount and the temporal Hour diff.

Our findings indicate that linear embeddings provide the best performance for numerical and temporal feature encoding regardless of the feature encoding scheme. However, considering that the Hour diff feature has a certain number of outliers and missing values in the dataset, we can observe that discretization of the real feature into intervals helps to solve the dataset consistency problems. Therefore, in our approach, we have chosen a linear encoding method for numerical and temporal features with their prior discretization.

### A.3 THE NECESSITY FOR A LANGUAGE MODEL

The utilisation of rich vector representations of event sequences is both beneficial and valuable in its own right. The use of such representations allows for the effective performance of different sequential models. In order to justify the necessity of using LLM in our approach, we conducted a comparison between the performance quality of our approach with a sequential model (GRU with 1024 hidden units) on top of event sequence encoder embeddings. This sequential model was fully

trained in the paradigm of our method to create vector representations of events. In order to ensure the effectiveness of our approach, we trained the model with the event encoder to solve each problem end-to-end.

The AlfaBattle 2.0 dataset was subjected to a series of comparative analyses in order to evaluate the efficacy of various models for predicting next transaction attributes. In these analyses, the sequential model was trained in a multi-task fashion with additional heads for each task. Furthermore, the AlfaBattle 2.0 and Age datasets were employed for the purposes of downstream problem solution. The results of this evaluation are presented in Table 10.

Table 10: The comparison results of the sequential model solution quality over rich event sequence representations with the ESQA approach. AB and Age denote tasks based on AlfaBattle 2.0 and Age datasets, respectively. The best results are highlighted in **bold**.

|  | Age | AB Default | AB MCC code | AB Amount | AB Hour diff |
|---|---|---|---|---|---|
|  | Accuracy | AUROC | Acc/F1 | MAE/MSE | MAE/MSE |
| Sequential model | 0.569 | 0.768 | 0.388/0.217 | 0.206/0.217 | 36.330/1699.2 |
| COLES | 0.640 | **0.792** | 0.440 / 0.351 | 0.197 / **0.082** | 36.05 / 1586.5 |
| ESQA | **0.699** | 0.757 | **0.546/0.546** | **0.191**/0.102 | **18.313/1033.9** |

A comparison of the quality metrics in Table 10 of the aforementioned approaches reveals that the sequential model with an event encoder trained in the ESQA paradigm does not demonstrate a comparable level of quality to the ESQA approach. Furthermore, the performance of this solution is inferior to that of one of the COLES baselines, which in turn represented the best approach in a range of typical event sequencing problems. Thus, the use of a language model significantly improves the quality of the approach, makes it possible to form tasks in natural language, and also enables the superior solution of several tasks at once.

## B IMPLEMENTATION DETAILS

### B.1 EVALUATION STRATEGY

We employed several classical machine learning metrics to thoroughly evaluate the proposed approach. As previously mentioned, ESQA is designed to handle tasks that can be framed as binary or multi-class classification as well as regression settings.

**Classification Metrics.** We utilised classification metrics for tasks that involved predicting a categorical feature of the next event or a characteristic of the entire sequence (e.g., default of a bank customer). For non-binary target tasks, we used Accuracy and F1-score. For binary target tasks, we employed the Area Under the Receiver Operating Characteristic curve (ROC-AUC). The model with the highest performance on these metrics was deemed the best.

To calculate the classification metrics Accuracy and F1 score using the language model's response in the question-answer format, we applied the following process. The question body was followed by an instruction specifying the format of the answer to clearly define the structure of the language model's output. The tokens predicted by the language model were then decoded into text, and the segments containing the desired answer were extracted. These extracted values $y$ were compared to the target $\hat{y}$ in a classification format, where the number of classes matched the cardinality of the predicted value. Subsequently, Accuracy and F1 were calculated as follows:

$$\texttt{Accuracy}(y, \hat{y}) = \frac{1}{n_{\text{samples}}} \sum_{i=0}^{n_{\text{samples}}-1} 1(\hat{y}_i = y_i) \tag{4}$$

$$\text{F1} = \frac{2 \cdot \text{TP}}{2 \cdot \text{TP} + \text{FP} + \text{FN}} \tag{5}$$

In this context, $TP$ represents the number of true positives, $FP$ stands for the number of false negatives and $FP$ denotes the number of false positives.

Table 11: Statistics of the datasets used for models evaluation.

| Dataset | AlfaBattle | Age | Gender | X5 | Taiwan |
|---|---|---|---|---|---|
| # events | 443 M. | 44 M. | 6,85 M. | 45,8 M. | 0.18 M |
| # sequences | 1,47 M. | 30 K. | 9,2 K. | 400 K. | 30 K. |
| Avg, seq. len. | 881.7 | 862.4 | 446.6 | 114.3 | 6 |
| # numeric | 3 | 1 | 3 | 3 | 3 |
| # categorical | 15 | 2 | 2 | 3 | 5 |
| # classes | 2 | 4 | 2 | 4 | 2 |
| train/val split % | 70/30 | 90/10 | 90/10 | 90/10 | 90/10 |

In calculating the ROC-AUC metric, we utilised the difference between the probabilities of the positive and negative response tokens.

**Regression Metrics.** To evaluate prediction performance for tasks with real-valued target variables, we employed Mean Absolute Error (MAE) and Mean Squared Error (MSE) metrics. For calculating these regression metrics, each question was accompanied by instructions specifying the format and range of the expected answer. The required numerical values (both real and integer) were then extracted from the LLM's textual predictions according to the given response structure. Instances where the prediction could not be interpreted as a number were excluded from the final metric calculation[1]. The selected numerical responses, denoted as $y$, were compared with the target values $\hat{y}$ for accurate assessment:

$$\mathrm{MAE}(y, \hat{y}) = \frac{1}{n_{\text{samples}}} \sum_{i=0}^{n_{\text{samples}}-1} |y_i - \hat{y}_i| \tag{6}$$

$$\mathrm{MSE}(y, \hat{y}) = \frac{1}{n_{\text{samples}}} \sum_{i=0}^{n_{\text{samples}}-1} (y_i - \hat{y}_i)^2 \tag{7}$$

### B.2 DETAILED DATASETS DESCRIPTION

A complete list of the datasets and a description of each dataset is given below. Main statistics and descriptions for each dataset are provided in Table 11. We took all datasets except Taiwan Dataset from Babaev (2024) repository, which provides data, preprocessing scripts and evaluation protocol. The Taiwan dataset we took from original source Yeh & hui Lien (2009) and then preprocessed with scripts from Babaev (2024).

**AlfaBattle2.0 dataset.** The AlfaBattle2.0 dataset Evgeny & Max (2021) consists of transaction activity records of bank customers over a two-year period, capturing spending, payments, and transfers. The primary goal is to estimate the probability of a customer defaulting on a loan within a given timeframe. The default rate in this dataset is 2.76%. Each customer is associated with a sequence of transactions, each described by 18 features: 3 numeric and 15 categorical. The numeric features include the normalized transaction amount, the number of hours since the customer's last transaction, and the number of days until the loan is disbursed. The categorical features encompass various identifiers: the merchant's code and category, the currency and type of payment card, and the city, country, etc. All categories are encoded with numeric values to ensure the dataset remains anonymized. The temporal component is defined by the attributes of hour, day of the week and week of the year, which in combination form the transaction date and time.

**Age Group Prediction Competition.** This dataset Sirius (2020) comprises anonymized transaction records of bank customers, with the aim of predicting the age group of each client based on their transactions. Each transaction is characterised by three features: a discrete MCC (Merchant Category Code) identifying the type of merchant, the transaction date, and the transaction amount. Transactions can be grouped according to the unique customer identifier specified in the transaction

---

[1]We made this assumption based on the rarity of such instances, given the clarity of the questions and the accompanying guidance provided for answering them.

description. The merchant identifier is also provided in text form, with categories such as 'book-shop', 'ATM', 'pharmacy', etc. This allows for a more detailed and nuanced analysis of spending patterns related to different age groups.

**Gender Prediction Competition.** The primary goal of this competition is to predict the gender of bank customers based on their transaction activity Max (2019). The dataset includes historical transaction and transfer data spanning one year and three months. Each transaction record is associated with a unique client ID and contains the time and date of the transaction, its type, the transaction amount, and a discrete identifier for the merchant point. The transaction amount is not normalised and can indicate both inflows and outflows of funds. A negative value signifies a debit, while a positive value denotes a credit to the account.

**Taiwan Default of Credit Card Clients.** This dataset Yeh & hui Lien (2009) includes customer transaction data from April to September 2005, and it is used to predict whether a customer will repay their borrowed credit. Each record in the dataset contains 8 real-valued attributes. Some attributes describe the customer's characteristics, such as age, education level, and marital status, while the remaining attributes provide details about the history of loan repayments.

**X5 Retail Hero: Uplift Modeling for Promotional Campaign.** Initially designed for an uplift modeling competition, this dataset focuses on predicting a customer's age based on their purchasing activity Babaev et al. (2022). Each purchase in the dataset is characterized by the time of the transaction, product type, segment, purchase amount, and the type of loyalty program associated with the customer.

### B.3 Baselines implementation details

Below, we provided details about the architectures and hyperparameters of the baseline approaches used in our study. While selecting baselines we tried to cover wide range of approaches for handling sequential data including contrastive, autoregressive, token masking and manual feature engineering and cover variety in architectures such as transformer or RNN.

**Handcrafted features with LightGBM**: This baseline aggregates numerical feature values across buckets of categorical features and includes statistics such as count, mean, variance, minimum, and maximum. The LightGBM classifier Ke et al. (2017) is then used for prediction. Such approach is following widely used practice of hand designing features.

**Randomly initialised RNN encoder**: This approach utilizes a randomly initialized and untrained RNN sequence encoder based on a unidirectional Gated Recurrent Unit (GRU) with a single hidden layer of size 1024. The resulting 1024-dimensional event sequence representations are used with LightGBM to solve the downstream task. This is a very simple baseline which main goal is to show whether any training is helpful for this task.

**NPPR** (Next Event Prediction and Past Reconstruction): The method proposed in Skalski et al. (2023) is based on two underlying pretraining tasks solving simultaniously 1) auto-regressive prediction of next events and 2) reconstruction of past of events with some predefined depth in time. This method demonstrated the best result on AlfaBattle and Age datasets so we included to our scope as a strong baseline.

**CoLES** (Contrastive Learning for Event Sequences): This method employs a self-supervised contrastive pretraining approach called CoLES Babaev et al. (2022) to generate vector representations of event sequences. The encoder is a recurrent neural network (RNN) GRU with one hidden layer of size 1024, producing final embeddings of the same size. A supervised classifier based on LightGBM is then trained using the pretrained embeddings. This is the one the strongest baseline demonstrated competitive quality on most task according to Babaev et al. (2022).

**CPC** (Contrastive Predictive Coding): This approach uses a similar sequence encoder architecture to CoLES but applies the Contrastive Predictive Coding (CPC) method van den Oord et al. (2018) for pretraining. CPC is a self-supervised technique for learning vector representations using an autoregressive model for non-discrete data sequences. This baseline represents common autoregressive type of pretraining and can be challenging basiline for next item prediction tasks.

The next series of baselines Barlow Twins, NSP, RTD, SOP are architecturally identical to COLES but differ in underlying pretraining task.

**Barlow Twins**: This method follows the same scheme and sequence encoder architecture as CoLES and CPC but implements a Barlow Twins Loss Zbontar et al. (2021) for encoder pre-training. Light-GBM is then used on the obtained embeddings for solving the downstream problem.

**NSP** (Next Sequence Prediction): This baseline employs an RNN sequence encoder with a unidi-rectional GRU and a single hidden layer of size 1024, pretrained on the Next Sequence Prediction task Devlin et al. (2019). The resulting 1024-dimensional embeddings are used with LightGBM for the downstream task.

**RTD** (Replaced Token Detection): Similar in architecture to the NSP baseline, this approach uses the Replaced Token Detection loss function from the ELECTRA paper Clark et al. (2020).

**SOP** (Sequences Order Prediction): Identical in architecture to NSP and RTD, this baseline uses the Sequences Order Prediction loss function from the ALBERT work Lan et al. (2019).

The next two baselines, TabFormer and TabGPT, utilize contemporary transformer neural network architecture.

**TabFormer**: This approach implements the TabFormer method Padhi et al. (2020), utilizing a Long-Former Beltagy et al. (2020) with 4 attention heads, 8 hidden layers of dimension 2048, and a max-imum of 2000 positions as the sequence encoder. The output embedding size is 2048. The encoder is pretrained using the Masked Language Modelling (MLM) task Devlin et al. (2019). LightGBM is then used on the obtained embeddings for solving the downstream problem.

**GPT**: This approach uses a GPT-2 architecture Radford et al. (2019) as the event sequence encoder, with 12 layers, 12 heads per layer, and position encoding up to 2056 positions. The embedding dimension is 768. The encoder is pretrained on an autoregressive task of predicting the fields of the next transaction, each using a separate head. LightGBM is used on the obtained embeddings for the downstream task.

**RNN with CoLES**: This baseline differs from the standard CoLES approach by adding several MLP heads to the event sequence encoder after contrastive pre-training. This architecture is then end-to-end fine-tuned on the target task after unsupervised contrastive pretraining with COLES. Fine-tuning have been achieved by setting smaller learning rate 0.00001 for RNN sequence encoder and large learning rate 0.001 for MLP heads.

**GPT with descr.**: This approach modifies the conventional GPT-2 baseline by applying discretiza-tion to the numerical features of events.

**CatBoost**: A simple implementation of the CatBoost algorithm Ostroumova et al. (2017) trained on event features. This baseline is applicable only for next item prediction task.

**Text LLM**: This text-based LLM approach serializes event features into a string using a template, selecting only the attributes necessary for the task while ignoring others due to the long token se-quence. The length of event sequences is also reduced to fit the language model's context. For this baseline, we used the FLAN-T5-xl Wei et al. (2021) model.

## C  SCALABILITY

Understanding the ability of the approach to scale to different sizes of data and base models, given the available computational resources, is an extremely important consideration. This is particularly true for models dealing with event sequences, as these can be very long and therefore require signif-icant model capacity.

In order to assess the scalability of our approach, we have evaluated the following two aspects of the inference and training processes:

- The inference time $T_{inf}$ of predicting a single data sample across event sequences of vary-ing lengths, from short ($l = 50$ events) to long ($l = 1500$ events).The observed timings are given in Table 12.

- The memory (peak vRAM) required for a training step using the Adam optimizer with a batch size of 1.

The measurements are summarised in Table 13. All measurements were conducted on the same 1-GPU NVIDIA A100-SXM4-80GB configuration and averaged over 10 steps. The components of the ESQA approach, apart from the LLM, were fixed: the Whisper-small architecture model was used as the transactional encoder and Q-Former-base as the connector. The -base/-large/-xl postfix in the ESQA configuration name indicates the size of the FLAN-T5 language model.

Table 12: Inference time estimation results of the ESQA approach with different baseline LLMs for dealing with event sequences of varying lengths. The best results are highlighted in **bold**.

| Configuration | $T_{inf}$, $l$=50, s. | $T_{inf}$, $l$=850, s. | $T_{inf}$, $l$=1500, s. |
|---|---|---|---|
| ESQA-xl | 0.122967 | 0.143583 | 0.144875 |
| ESQA-large | 0.118058 | 0.132152 | 0.137945 |
| ESQA-base | 0.071870 | 0.105394 | 0.126081 |

Table 13: The results of measuring the amount of memory that is utilised during different parts of a single training step for the ESQA approach.

| Approach | Total Size, Gb. | Grad. calc., Gb. | Backward pass, Gb. | Optim. step, Gb. |
|---|---|---|---|---|
| ESQA-xl | 11.62 | 11.62 | 23.24 | 46.47 |
| ESQA-large | 3.89 | 3.89 | 7.77 | 15.54 |
| ESQA-base | 1.88 | 1.88 | 3.75 | 7.51 |

As a result of estimating the time required to run ESQA on sequences of different lengths, it can be seen that the inference time does not grow significantly with increasing sequence length ( considering the variability of values within runs of the same model configuration), which demonstrates the scalability of the approach to event sequences of varying durations.

The length of the event sequence impacts only on the encoder inference time, because after encoding sequence to embeddings we transform them to the fixed number of queries with Q-Fomer connector. So afterwards LLM takes a small and fixed number of embeddings for the whole event sequence, no matter the initial event sequence size.

A higher increase in the inference time as well as in the memory requirements for training is observed when the baseline LLM becomes larger, which leads to a careful selection of this component of the approach depending on the resources available.

