# OpenReview forum: "ESQA: Event Sequences Question Answering"
_ICLR.cc/2025/Conference — ICLR 2025 Conference Withdrawn Submission_

### Official Review · Reviewer_Jrbh · 2024-11-02

**Soundness:** 3
**Presentation:** 2
**Contribution:** 3
**Rating:** 6
**Confidence:** 4

**Summary:**

This paper presents a novel approach to handling event sequences (ESs) by adapting LLMs to the ES domain for multiple downstream tasks like extraction and prediction, without the need for extensive fine-tuning. This work addresses challenges in processing ESs, including long sequence lengths and temporal features. Key contributions include: (1) proposing a novel framework for processing long ESs using LLMs, and (2) leveraging the Q-Former to efficiently compress and integrate ESs into LLMs. State-of-the-art results on multiple ES tasks across several datasets are achieved.

**Strengths:**

- Originality: The paper introduces a creative adaptation of LLMs to the ES domain that has not been extensively explored with such models. While prior work has adapted LLMs to other non-language data (e.g., time series), the specific application to ESs with the question-answering format is novel.
- Quality: The paper offers extensive experiments on relevant datasets from real-world domains. The use of LLMs for ES modeling and the integration of Q-Former as a connector between the LLM and event sequence encoder are sound technical contributions.
- Clarity: The paper is generally clear in its description of the methods and experiments.
- Significance: The proposed approach is highly significant in the context of applied machine learning, particularly in industries reliant on ES data. The ability to leverage LLMs for ES tasks with minimal fine-tuning could lead to substantial efficiency improvements in data processing and decision-making systems.

**Weaknesses:**

- Quality: The proposed method has certain limitations, particularly in handling imbalanced classes and discretization errors in numeric features, as noted by the authors.
- Clarity: There are several typos and styling issues, for example:
  - Numerous citations should be in parenthesis using \citep command.
  - The authors have mentioned that "the second best results are underlined" in Table 4's caption, but the table contains no underlines.
  - In the captions of Table 3-5, "baseline approaches presented in Section 4.2.1" should actually be Section 4.1 or Appendix B.3.
- Significance: The model has been tested primarily on financial data, but not on more diverse domains, such as healthcare or social media interactions, thus the domain generalizability of the model is not demonstrated.

**Questions:**

1. What is the compress rate between the query embeddings and the textual representation of the structured data?
2. Have the authors considered alternative approaches for dealing with imbalanced classes, such as integrating class weights into the cross-entropy loss function?

---

> ### Author Response · Authors · 2024-11-26
>
> We sincerely thank you for your thoughtful analysis and valuable comments on our work. Your feedback has been instrumental in identifying areas for improvement, and we will incorporate your suggestions during the revision process. Below, we address your questions in detail:
>
> ### 1. Compression Rate and Dimensionality Reduction
>
> The compression rate between the textual representation of an event sequence and its reduced dimensionality after the Q-Former (query size) primarily depends on:
>
> - The number and description of attributes used during event serialization.
>
> - The tokenization technique employed by the chosen LLM.
>
> For the AlfaBattle dataset, we measured these rates using two LLMs with different tokenization methods while including all event features (excluding numeric client and transaction IDs). Our results demonstrate that ESQA achieves significant context compression advantages over approaches relying on text serialization of event sequences.
>
> | Dataset        | Model           | Sequence length (in #events) | Sequence length l1 (in tokens), mean +- std | #queries in ESQA l2 | Compression ratio (as l1/l2) |
> |----------------|-----------------|------------------------------|---------------------------------------------|---------------------|------------------------------|
> | AlfaBattle 2.0 | FLAN-T5-small   | 278 +- 241                   | 25 580 +- 22 202                            | 32                  | x799                         |
> | AlfaBattle 2.0 | Mistral-7b-v0.1 | 278 +- 241                   | 34 756 +- 30 166                            | 32                  | x1018                        |
>
> ### 2. Addressing Class Imbalance
>
> We tested additional modifications to our approach to address class imbalance, including:
>
> - Training an auxiliary classifier.
> - Applying weights in the cross-entropy loss function.
>
> | Model                | Default prediction, AUC |
> |----------------------|-------------------------|
> | ESQA with classifier | 0.7434                  |
> | ESQA                 | 0.7568                  |
>
> When evaluated on the challenging task of client default prediction, which is highly imbalanced (97% negative vs. 3% positive), we found that these modifications were unfortunately insufficient to effectively address the issue of unbalanced target classes in datasets with significant class imbalance.
>
> We greatly appreciate your insightful suggestions and corrections, which will guide us in improving the clarity and robustness of our manuscript. Thank you for your support in enhancing the quality of our work.

---

### Official Review · Reviewer_KKFh · 2024-11-03

**Soundness:** 2
**Presentation:** 3
**Contribution:** 2
**Rating:** 3
**Confidence:** 4

**Summary:**

This paper introduces an LLM-based method for modeling event sequences. By leveraging LLMs, this research aims to improve the generalization and modeling quality of the existing methods for event sequence prediction tasks. Benefiting from LLM's knowledge from other domains, the proposed method is able to perform unseen tasks. The experimental results demonstrate the capability of the proposed method in achieving state-of-the-art results on certain tasks compared to the existing methods.

**Strengths:**

1. The paper proposes a method to encode features of different formats and integrate the embedded features with questions as input to LLMs. The motivation for using LLMs is clearly presented.
2. By leveraging LLMs, the proposed method can generalise to unseen tasks.
3. The proposed method outperforms the baseline methods on some of the datasets.
4. In general, the paper is well-written and easy to follow.

**Weaknesses:**

**Novelty**:
1. The paper claimed it is the first to apply LLMs to event sequences. This claim does not seem to be correct. There have studies [1,2] using LLMs for event sequences.
2. The technical novelty of this paper is weak. The event encoder adopts several well-known machine learning techniques, such as binning, to convert the raw features to learnable embeddings. The connector is a direct application of an existing technique Q-former. The training protocol applies LORA, a well-known fine-tuning technique for LLMs. It is hard to see the originality of the proposed method.

**Method Design**:

1. Although discretisation is one way to convert numeric features into categorical features, it also increases the dimensions and parameters of the model. If an event has a large number of features, the event embedding (obtained by concatenating the embedding of each feature) can be very large, which may lead to a large number of parameters in the projection layer. Discretisation may not be parameter-efficient in this case.
2. The method is limited to categorical and numeric data. It lacks support for other data types. For instance, in social media, plain text and images are very common attributes.
3. The motivation for choosing only FLAN-T5 as the backbone is unclear.

**Experiments**:

1. Although the paper claimed event sequences are prevalent in real-world applications, including social networks and healthcare, the paper only evaluates the methods on fintech data.
2. The latest methods compared in the paper were published in 2022. More recent methods could be included as baselines, such as [1,2]. Besides, the proposed method does not have fundamental differences compared to LLMs for time series or temporal knowledge graphs, except for the feature encoder; it is a question why the authors do not compare with those LLM-based methods in other domains.
3. The proposed method underperforms the baselines on certain datasets. However, there is no in-depth analysis of the reason.
4. LLMs are known to be unstable in text generating. The output of independent runs might be different. However, there is no standard deviation is reported in the experimental results. Compared to existing methods, LLM-based method might have a larger variance. The authors need to justify that the proposed method is stable and that the improvements are statistically significant.
5. The experimental setting for the generalization abilities is not clearly presented. The method ESQA m/t is trained in a multi-task setting. Do these tasks include the task in the testing phase? The low performance of ESQA z/s is not surprising because it is trained only one a single task.

**Minor comments**:

1. The citation format seems to be incorrect. For instance, "This type of data is widely spread in domains like geoscience Bergen et al. (2019)" --> "This type of data is widely spread in domains like geoscience (Bergen et al. 2019)".
2. The definition $c_j^{cat} \in |c_{j}|$ is confusing. $|c_{j}|$ is often used to denote the size of a set.
3. On page 3, "a sequence of f-dimensional vectors in$E_{\theta}$". There should be spacing between "in" and "$E_{\theta}$".

**Reference**:

[1] Xiaoming Shi, Siqiao Xue, Kangrui Wang, Fan Zhou, James Zhang, Jun Zhou, Chenhao Tan, Hongyuan Mei. Language Models Can Improve Event Prediction by Few-Shot Abductive Reasoning. NeurIPS 2023.

[2] Matthew McDermott, Bret Nestor, Peniel Argaw, Isaac S Kohane. Event Stream GPT: A Data Pre-processing and Modeling Library for Generative, Pre-trained Transformers over Continuous-time Sequences of Complex Events. NeurIPS 2024.

**Questions:**

1. Converting the structured features of an event is not difficult. Once converted, it becomes a sequence of embedding, which is identical to other LLM-based sequence modeling methods, e.g., those for time series. What is the uniqueness of this paper compared to other LLM-based sequence prediction problems?

2. What is the reason for only considering numeric and categorical features?

3. Why FLAN-T5 is used as the backbone? Have you tested other LLMs such as GPT4, Llamma3, etc.?

4. Only evaluating the method on fintech data cannot justify its generalisability. Why not compare these methods on social media data and healthcare data?

5. What is the exact reason that the proposed method underperforms the competitors on AlfaBattle and Gender datasets? What are the unique properties of these datasets that lead to the performance degrade of the method? You mentioned the two datasets exhibit class imbalance. Does this mean your method is vulnerable to imbalanced data?

6. What are the tasks used for training ESQA m/t, and how are they different from the tasks in the testing phase?

---

> ### Author Response · Authors · 2024-11-26
>
> We sincerely thank you for your detailed and thoughtful feedback. Your comments and suggestions have greatly helped us refine and improve our manuscript. Below, we address your concerns point by point.
>
> **Novelty**. The LLM-based method [1] you referenced addresses problems distinct from those tackled by our solution. Specifically, [1] is limited to predicting only the time and type of events, whereas our approach handles more complex scenarios. Similarly, while [2] addresses a related problem, it focuses on a domain with significantly different data dynamics.
>
> **Experiment design**. Thank you for the clarification. We acknowledge the variability in the results of LLM-based approaches due to the specificity of text response generation. We provide an extended table of the main results with standard deviations (extended Table 3).
>
> | Dataset | AlfaBattle | Age | Gender | X5 | Taiwan |
> |--|--|--|--|--|--|
> | Metric | AUCROC | Accuracy | AUCROC | Accuracy | AUCROC |
> | handcrafted feat. | 0.7792 ± 0.0006 | 0.629 ± 0.002 | 0.877 ± 0.010 | 0.547 ± 0.001 | 0.784 |
> | Randomly init. RNN | 0.6456 ± 0.0009 | 0.375 ± 0.003 | 0.593 ± 0.020 | | 0.722 |
> | CPC | 0.7919 ± 0.0004 | 0.602 ± 0.004 | 0.851 ± 0.006 | 0.525 ± 0.001 | - |
> | Barlow Twins | 0.7878 ± 0.0009 | 0.634 ± 0.003 | 0.865 ± 0.007 | | - |
> | CoLES | 0.7921 ± 0.0003 | 0.640 ± 0.004 | 0.881 ± 0.006 | 0.539 ± 0.001 | - |
> | NSP | 0.7655 ± 0.0006 | 0.621 ± 0.005 | 0.852 ± 0.011 | 0.425 ± 0.002 | - |
> | RTD | 0.7910 ± 0.0006 | 0.631 ± 0.006 | 0.855 ± 0.008 | 0.520 ± 0.001 | - |
> | SOP | 0.7238 ± 0.0010 | 0.512 ± 0.002 | 0.785 ± 0.007 | 0.428 ± 0.001 | - |
> | TabFormer | 0.7862 ± 0.0042 | 0.580 | 0.828 | 0.393 | 0.697 |
> | GPT | 0.7737 ± 0.0032 | 0.574 | 0.785 | 0.511 | 0.732 |
> | NPPR | 0.7980 | 0.642 | - | - | - |
> | ESQA (ours) | 0.7568 ± 0.05 | 0.699 ± 0.013 | 0.850 ± 0.004 | 0.598 ± 0.01 | 0.793 ± 0.008 |
>
> Furthermore, we hypothesise that the main reason why ESQA shows a lower quality than the proposed baselines in some of the tasks is the imbalance of the target variable. This problem is most evident in downstream tasks concerning LLM.
>
> 1. **Challenges with Structured Data in LLMs**
>
> We respectfully disagree with the assertion that converting structured event features into an LLM-friendly format is straightforward. Existing time series methods generally handle only numerical and time-based features, whereas our approach integrates diverse feature types, including categorical and textual attributes, into a unified vector representation.
>
> 2. **Feature Types and Domain-Specific Limitations**
>
> Our current focus on numerical and categorical features reflects the characteristics of financial data, where these types dominate. However, our approach is flexible and can be extended to accommodate other event attributes in future work.
>
> Regarding numeric attribute embeddings, while this increases dimensionality, it enables more effective encoding of numeric features (as shown in Section A 2.1, Table 9). Dimensionality is reduced at the event encoder stage to ensure the model's efficiency.
>
> 3. **Model Choice and Comparison**
>
> We selected the T5 family for its encoder-decoder architecture, which enhances contextual representation of long event sequences. Our experiments demonstrated that FLAN-T5-small outperformed Galactica (125M parameters) in tasks involving numeric attributes, such as determining the most frequent value range or average value on the AlfaBattle dataset.
>
> | | Accuracy (multi-class, weighted) |
> |-----------------------|----------------------------------|
> | FLAN-T5-small (80 M.) | 0.377 |
> | Galactica-125 M. | 0.312 |
>
> Additionally, FLAN-T5 achieved lower MAE compared to decoder-only models for similar tasks.
> |                       | MSE    | MAE    |
> |-----------------------|--------|--------|
> | FLAN-T5-small (80 M.) | 0.0084 | 0.0611 |
> | Galactica-125 M.      | 0.0129 | 0.0878 |
> | GPT-Neo-125 M.        | 0.0081 | 0.067  |
>
> We plan to extend our experiments to newer decoder models, including those from the Mistral, LLaMA, and Qwen families.
>
> However, the transition to decoder-only models requires significant modification of the current version of the method and its components. Although the mechanisms and properties of decoder-only models make them suitable for predictive tasks, encoder-decoder LLMs encode event sequences more efficiently due to the encoder part.

---

> > ### Author Response · Authors · 2024-11-26
> >
> > 4. **Domain and Future Applicability**
> >
> > Our validation focused on a domain rich in event sequences (lines 309–313). However, the approach is not domain-restricted, and we intend to explore its application to other domains as part of future research.
> >
> > 5. **Class Imbalance and Training Techniques**
> >
> > Class imbalance in downstream tasks is a well-documented challenge in event sequence datasets. We currently mitigate this using next-token prediction loss and dataset resampling. While methods like Focal loss with a classification head could further address this issue, such approaches reduce model flexibility and fall beyond our current scope.
> >
> > 6. **Multitask Training Setup**
> >
> > Task selection for multitask training varied based on experimental goals:
> >
> > - For evaluating prediction accuracy, pre-training was focused on next-event feature prediction tasks, tested on similar tasks.
> >
> > - For assessing generalization, training involved contextual tasks (e.g., feature trends and statistics), followed by testing on next-event prediction tasks.
> >
> > Once again, we thank you for your insightful comments. We have also incorporated your corrections into the manuscript to improve its overall readability and quality.

---

> > > ### Comment · Reviewer_KKFh · 2024-11-27
> > >
> > > Dear authors,
> > >
> > > Thank you for your response! After reading your rebuttal, I have the following questions:
> > > 1. You mentioned your method "handles more complex scenarios" compared to [1]. Could you please clarify what those scenarios are and why [1] can't be easily adapted to those scenarios? A more thorough comparison may help to highlight your novelty.
> > > 2. Regarding the challenges with structured data in LLMs, could you provide more details of the technical difficulty? I understand that time-series methods only handle numeric data. However, what makes it difficult to add categorical data to the prompt?

---

### Official Review · Reviewer_cE6Z · 2024-11-03

**Soundness:** 3
**Presentation:** 2
**Contribution:** 2
**Rating:** 3
**Confidence:** 4

**Summary:**

This paper introduces ESQA (Event Sequences Question Answering), a framework designed to handle question-answering tasks in the domain of event sequences. Leveraging large language models (LLMs), ESQA addresses unique challenges in processing temporally structured data, such as managing long input sequences and encoding time and numeric features effectively. The framework showcases its adaptability across various downstream tasks without extensive fine-tuning, achieving comparable performance across multiple event-sequence datasets in domains like finance and e-commerce.

**Strengths:**

S1: Experimental Design: The paper supports its proposed approach with a series of experimental setup, utilizing multiple datasets that span various event sequence types and domains, enhancing the robustness and applicability of the findings.

S2: Each component of the ESQA framework is backed by experimental results, validating the selection and effectiveness of different modules.

S3: ESQA achieves state-of-the-art or competitive results on a range of tasks, highlighting its versatility and effectiveness in handling diverse event-sequence tasks.

**Weaknesses:**

Overall, the paper lacks novelty beyond the designed QA framework with backbone encoder and LLM. Moreover, the organization and experiment settings in paper can be further optimized.

W1: Inadequate Background Explanation: The paper contains some strongly worded statements in the background, such as “no effort was made to adapt LLMs to event sequences.” However, event sequence tasks share similarities with event prediction tasks, making it essential for the authors to clarify the differences between these two. Since LLMs have already been applied to various event prediction tasks, it would be valuable to consider these models as baselines for comparison. Additionally, there is considerable prior work on LLMs in question answering, and including these models as baselines could further strengthen the experimental design.

W2: Lack of Rigorous Annotation: The methodology section lacks precise annotation and contains numerous typographical issues. For example, “Q-Former” in the related work section is not properly cited, which could lead to misunderstandings. Some symbols, such as special tokens, are only briefly introduced in figures without corresponding explanations in the text. This could create confusion among readers, highlighting the need for careful annotation and early definition of terms.

W3: Insufficient Justification of Component Selection: The paper could enhance the methodology by explaining the motivations behind selecting each component of the framework, beyond the experimental advantages. A discussion of why each component was chosen in alignment with the framework’s goals would strengthen the readers' understanding of the architecture.

W4: Mismatch Between Methodology Text and Figures: Certain descriptions in the methodology section do not fully align with the figures. For example, how the Input prompt, query, and Question are integrated is not clearly explained, and the division in the embedding matrices shown in Figure 2 could lead to misinterpretation. Moreover, although the authors state that the LLM is not fine-tuned, LoRA, a traditional fine-tuning technique—is used. These inconsistencies should be clarified to avoid potential misunderstandings.

W5: Limited Experimental Baselines: The experimental design could benefit from considering more comprehensive baseline models from existing LLM research, especially LLM applications in question answering and traditional event prediction, to provide a more robust evaluation of the proposed framework.

**Questions:**

Refer to the weakness.

---

> ### Author Response · Authors · 2024-11-26
>
> Thank you for your valuable feedback. Below, we address the points you raised:
>
> 1. **Related Work**
>
> We appreciate your analysis and will incorporate your recommendations in the revised paper. The updated version will provide a more detailed discussion of related approaches in event prediction tasks and LLM-based question answering.
>
> 2. **Special Tokens**
>
> Thank you for your comments on improving clarity. Our approach uses three special tokens:
>
> - `<trx>` and `</trx>`: Mark the start and end of event sequence tokens, enabling the LLM to distinguish between injected modalities and textual embeddings.
>
> - **Attribute Token**: An optional token added to instructions to guide the model’s focus on key attributes within event sequences. These tokens are added to the model dictionary and trained during the dataset's training phase.
>
> 3. **Design choices**
>
> In addition to empirical choices in favour of one or the other component, we were also guided by the following basic idea: feeding raw long event sequences significantly increases the length of the LLM context and leads to an inefficient use of useful information (the effect known as Needle in a Haystack), despite the possibility of context expansion in modern LLMs.
>
> As a consequence, we used a separate encoder model to represent each individual event in a time-domain vector representation. In this key, models based on the Transformer decoder architecture seemed promising. Furthermore, due to the similarity of event sequences and other time-dependent data types (e.g. audio), we wanted to use the capabilities of specialised encoders for these modalities for ES analysis. A more detailed description of the event encoder modality selection process is given in Appendix A1.
>
> The further reduction of the dimensionality of the event sequence embedding vector was a key point in realising a context-aware and trainable method, which the Q-Former module turned out to be. The detailed selection of the connection layer is described in Appendix A2.
>
>
>
> 4. **Methodology Details**
>
> We will refine the methodology description to address your concerns:
>
> - Prompt and question tokens are encoded by the LLM and concatenated with the reduced event sequence embeddings after the Q-Former (Figure 1).
>
> - In Figure 2, the color-coded embedding matrix highlights the independent encoding of each event feature.
>
> - The LoRA PEFT method in our approach differs from classical fine-tuning, as it modifies only low-rank adapter layers, requiring fewer computational resources.
>
> 5. **Baseline Comparisons**
>
> Regarding extending the number of baselines: in the transactional domain of event sequences, the selected baselines remain SOTA for the tasks. Therefore, comparing ESQA with them provides the most relevant evaluation.

---

### Official Review · Reviewer_Y9kg · 2024-11-03

**Soundness:** 2
**Presentation:** 2
**Contribution:** 2
**Rating:** 5
**Confidence:** 4

**Summary:**

The authors present a new approach to apply large language models to the task of event sequence prediction. This approach is a little fine-tuned and less expensive to use. It takes into account both time and numerical sequence processing. The authors chose FLAN-T5 as the base model and conducted experiments on several datasets related to customer transactions. In addition to experiments on specific downstream tasks, the generalization ability of the large language model for event sequence prediction was tested.

**Strengths:**

1. The authors take advantage of categorical and numeric features from tabular neural networks at the same time in Section 3.2.
2. In Section 3.3, the choice of Whisper as an Encoder after extensive comparative experiments is convincing.

**Weaknesses:**

1. I think the biggest problem with this paper is the lack of sensible baselines. The authors ignore some recent applications of large models for event prediction in the last year. For example [1], [2]. The authors use earlier baselines and therefore the results obtained are not convincing. In particular, according to Table 3, ESQA does not achieve a consistent lead compared to NPPR either.
[1] Analyzing Temporal Complex Events with Large Language Models? A Benchmark towards Temporal, Long Context Understanding
[2] MIRAI: Evaluating LLM Agents for Event Forecasting

2. I think the novelty of the ESQA method itself is limited. The authors have mentioned time-series prediction in the related work section. In fact, there is a lot of recent work dealing with complex data structures such as time-series, e.g., [3], [4]. They have embedded complex data structures individually and used the combination of freezing and tuning. So, I don't think similar ideas used for event sequence prediction task is very novel.
[3] TIME-LLM: TIME SERIES FORECASTING BY REPROGRAMMING LARGE LANGUAGE MODELS
[4] LLM4TS: Aligning Pre-Trained LLMs as Data-Efficient Time-Series Forecasters

3. The authors' descriptions of their methods and the results are often confusing. For example, the Encoder model is used in Figure 1, and the Event Sequence Encoder is used in Figure 3.

**Questions:**

1. Why did you choose only a few datasets for transactions and not for other domains? You have mentioned in the introduction section that event prediction is widely used in various fields such as healthcare, sociology etc. I have read the sources of the datasets you have used and found that they are rarely used by other works. So, I think the datasets are not representative.
2. Why are all the large language models you use for experiments the FLAN-T5 series? Why don't you use the newer LLaMA series or the Mistral series?
3. How did you get the two empirical values of 0.6 and 1.56 in rows 168-169? Have you tried other values?

---

> ### Author Response · Authors · 2024-11-26
>
> We appreciate your detailed feedback and have carefully addressed the points raised:
>
> 1. **Novelty and Baseline Comparisons**
>
> While we acknowledge your concern about the novelty of the baselines, we emphasize that in the transactional domain of event sequences used for validation, these baselines remain state-of-the-art (SOTA) for the tasks. Comparing ESQA with these methods is therefore essential for fair evaluation.
>
> The LLM-based approaches you referenced address fundamentally different data types. For example:
>
> - In [1], *Temporal Complex Events* are defined as semantically related textual articles, unlike our Event Sequences, which involve diverse attributes (e.g., numbers, categories, text).
>
> - In [2], events are constrained to predefined attributes describing specific interactions (e.g., international relations), whereas our sequences have more varied structures.
>
> 2. **Background Papers**
>
> Thank you for sharing useful references. We recognize the established applications of similar techniques to multivariate time series. However, their limitations become evident when handling more complex datasets with non-uniform time intervals between events, underscoring the need for our approach.
>
> 3. Thank you for your comment, we have noted it and will certainly take it into account to improve our work.
>
>
>
> In response to your questions:
>
> 1. **Domain specificity**
>
> Regarding validation, our focus was on a domain rich in diverse event sequences (lines 309-313), but this does not preclude broader applicability. Future work will extend our experiments to other domains.
>
> 2. **Model Selection:**
>
> The T5 family was chosen for its encoder-decoder architecture, which excels at producing contextual representations for long event sequences. Additional experiments also showed that FLAN-T5-small outperformed Galactica (125M parameters) on tasks involving numerical features, such as determining frequent value ranges and mean values, as tested on the AlfaBattle dataset.
>
>  |                       | Accuracy (multi-class, weighted) |
> |-----------------------|:--------------------------------:|
> | FLAN-T5-small (80 M.) |               0.377              |
> | Galactica-125 M.      |               0.312              |
>
> Moreover, FLAN-T5 achieved better MAE than comparable decoder-only models for similar tasks.
>
> |                       | MSE    | MAE    |
> |-----------------------|--------|--------|
> | FLAN-T5-small (80 M.) | 0.0084 | 0.0611 |
> | Galactica-125 M.      | 0.0129 | 0.0878 |
> | GPT-Neo-125 M.        | 0.0081 | 0.067  |
>
> We plan further experiments with newer models (Mistral, LLaMA, Qwen, etc.).
>
> However, the transition to decoder-only models requires significant modification of the current version of the method and its components. Although the mechanisms and properties of decoder-only models make them suitable for predictive tasks, encoder-decoder LLMs encode event sequences more efficiently due to the encoder part.
>
> 3. **Feature Embedding Size:**
>
> The chosen embedding sizes for categorical features were determined based on experiments across two datasets (AlfaBattle, Age), consistently yielding the best downstream task performance.
>
> Thank you for your thoughtful comments, which have guided us in improving this work.

---

> > ### Comment · Reviewer_Y9kg · 2024-12-02
> >
> > thank you for your explanation. I will keep my original scores.

---

### Note · Authors · 2024-12-27

I have read and agree with the venue's withdrawal policy on behalf of myself and my co-authors.